# Endovascular thrombectomy in acute stroke with a large ischemic core: A systematic review and meta-analysis of randomized controlled trials

Chun-Hsien Lin [1], Meng Lee [2]*, Bruce Ovbiagele [3], David S. Liebeskind [4], Borja Sanz-Cuesta [5], Jeffrey L. Saver [4]

1 Division of Neurology, Department of Internal Medicine, Ditmanson Medical Foundation, Chia-Yi Christian Hospital, Chiayi, Taiwan, 2 Department of Neurology, Chang Gung University College of Medicine, Chang Gung Memorial Hospital, Chiayi branch, Puzi, Taiwan, 3 Department of Neurology, University of California, San Francisco, California, United States of America, 4 Comprehensive Stroke Center and Department of Neurology, University of California, Los Angeles, California, United States of America, 5 Department of Neurology, Hofstra University and Northwell Health, New York, New York, United States of America

* menglee5126@gmail.com

## Abstract

### Background

Endovascular thrombectomy (EVT) is the standard treatment for acute ischemic stroke due to internal carotid artery (ICA) or middle cerebral artery (MCA) M1 occlusion with a small ischemic core. However, the effect of EVT on acute stroke with a large ischemic core remains unclear. This study aimed to evaluate the association of EVT plus medical care versus medical care alone with outcomes in patients with acute stroke and a large ischemic core due to ICA or MCA M1 occlusion.

### Methods and findings

PubMed, the Cochrane Central Register of Controlled Trials, and ClinicalTrials.gov were searched from January 1, 2000 to September 25, 2024. There were no language restrictions. Randomized controlled trials (RCTs) of patients with acute stroke and a large ischemic core that compared EVT plus medical care versus medical care alone were evaluated. We computed the random-effects estimate based on the inverse variance method. Risk ratio (RR) with 95% confidence interval (CI) was used to measure outcomes of EVT plus medical care versus medical care alone. The primary outcome was functional independence, defined as modified Rankin Scale (mRS) of 0–2 at 90 days post-stroke; and the lead secondary outcome was reduced disability, defined as ordinal shift of mRS. Safety outcomes were requiring constant care or death (mRS 5–6), death, and early symptomatic intracranial hemorrhage (sICH). Grading of Recommendations Assessment, Development and Evaluations (GRADE) was used to evaluate summaries of evidence for the outcomes. We

**Data availability statement:** All relevant data is available within the manuscript and Supporting Information files.

**Funding:** The author(s) received no specific funding for this work.

**Competing interests:** The authors have declared that no competing interests exist.

**Abbreviations:** ACR, assumed comparator risk; CI, confidence interval; CT, computed tomography; DWI, diffusion-weighted imaging; EVT, endovascular thrombectomy; GRADE, Grading of Recommendations Assessment, Development and Evaluations; ICA, internal carotid artery; MCA, middle cerebral artery; MRI, magnetic resonance imaging; mRS, modified Rankin Scale; NNT, number needed to treat; PRISMA, Preferred Reporting Items for Systematic Reviews and Meta-Analyses; RCT, randomized controlled trial; RR, risk ratio; sICH, symptomatic intracranial hemorrhage.

included six RCTs comprising 1870 patients (826 females [44.2%]) with acute stroke and a larger moderate or large ischemic core due to ICA or MCA M1 occlusion. All patients were nondisabled before stroke. Pooled results showed that at 90 days post-stroke, EVT plus medical care, compared with medical care alone, was associated with greater functional independence (RR 2.53, 95% CI [1.95, 3.29]; $p < 0.001$; number needed to treat [NNT], 9, 95% CI [6,15]) and reduced disability (common odds ratio 1.63, 95% CI [1.38, 1.93]; $p < 0.001$; NNT, 4 [minimum possible NNT, 2; maximum possible NNT, 6]). EVT plus medical care, compared with medical care alone, was associated with a lower risk of requiring constant care or death (RR 0.74, 95% CI [0.66, 0.84]; $p < 0.001$; NNT, 7, 95% CI [6,11]). EVT plus medical care, compared with medical care alone, was associated with a nonsignificantly higher proportion of patients with early symptomatic intracranial hemorrhage (RR 1.65, 95% CI [1.00, 2.70]; $p = 0.05$). The rates of death were not significantly different between the EVT plus medical care and medical care alone groups (RR 0.86, 95% CI [0.72, 1.02]; $p = 0.08$). Main limitations include variability in imaging definitions of large core and inclusion of both larger moderate and large cores in the analysis.

## Conclusions

Among patients with acute stroke and a larger moderate or large ischemic core due to ICA or MCA M1 occlusion who were nondisabled before stroke, EVT plus medical care, compared with medical care alone, may be associated with improved functional independence, reduced disability, and reduced rates of severe disability or death at 90 days post-stroke.

PROSPERO registration number: CRD42024514605

## Author summary

### Why was this study done?

- Endovascular thrombectomy (EVT) plus medical care is more effective than medical care alone in improving functional independence among patients with acute stroke and a large ischemic core due to internal carotid artery (ICA) or middle cerebral artery (MCA) M1 occlusion.

- Whether EVT is beneficial for acute stroke with a large ischemic core is not well-settled.

### What did the researchers do and find?

- We conducted a meta-analysis including six relevant randomized controlled trials comprising 1,870 patients with acute stroke and a large ischemic core due to ICA or MCA M1 occlusion who were nondisabled prior to stroke, comparing EVT plus medical care with medical care alone.

- EVT plus medical care was associated with a higher rate of functional independence at 90 days post-stroke compared to medical care alone (20% versus 7%).

- EVT plus medical care was also associated with a lower rate of requiring constant care or death (43% versus 58%).

## What do these findings mean?

- Among patients with acute stroke and a large ischemic core who were nondisabled prior to stroke, EVT may provide clinical benefit.

- These findings support the ethical and clinical justification for the use of EVT plus medical care in patients with a large ischemic core due to ICA or MCA M1 occlusion.

- A main limitation is that this analysis includes both patients with larger moderate and true large ischemic cores, rather than patients with strictly large cores alone.

## Introduction

Endovascular thrombectomy (EVT) plus medical care has been proven more effective than medical care alone in improving functional independence and reducing disability in patients with acute ischemic stroke due to internal carotid artery (ICA) or middle cerebral artery (MCA) M1 occlusion and a small to moderate ischemic core, defined by an Alberta Stroke Program Early Computed Tomographic Score (ASPECTS) of at least 6 or an ischemic core volume ranging from less than 31 ml to less than 70 ml, depending on the study criteria [1–3]. EVT is a minimally invasive procedure in which a catheter is inserted into a large artery, typically in the groin, and advanced to the blocked brain artery to remove the clot, restoring blood flow [1]. However, approximately 20% of patients with acute ischemic stroke due to ICA or MCA M1 occlusion have a large ischemic core, and these patients generally have a poor prognosis, including stroke progression, dependence on constant care, and death [4].

Most randomized controlled trials (RCTs) enrolling patients with a large ischemic core indicated a benefit in functional outcome after EVT plus medical care compared with medical care alone [5–9]. However, one trial did not show significant benefits upon functional outcome [10]. Meta-analysis of RCTs may help to resolve conflicting results of individual trials, increase precision of effect magnitude estimates, enhance generalizability, and inform clinical practice guidelines [11]. We therefore conducted a systematic review of RCTs to evaluate the association of EVT plus medical care versus medical care alone with outcomes in patients with acute stroke and a large ischemic core due to ICA or MCA M1 occlusion.

## Methods

This study is reported as per the Preferred Reporting Items for Systematic Reviews and Meta-Analyses (PRISMA) guideline (S1 Checklist) [12]. The protocol was registered with PROSPERO (CRD42024514605).

### Search methods and resources

We searched PubMed, the Cochrane Central Register of Controlled Trials, and the clinical trial registry maintained at ClinicalTrials.gov for studies published from January 1, 2000 (the first modern thrombectomy device was induced in early 2000s), to September 25, 2024 using the following terms: stroke or cerebrovascular disease or brain ischemia or brain infarct or cerebrovascular accident and large core or large infarct or large ischemic and endovascular therapy or endovascular thrombectomy or endovascular treatment or mechanical thrombectomy or intra-arterial therapy. We restricted

our search to RCTs. There were no language restrictions. We also reviewed the introduction and discussion sections of retrieved trials to identify additional trials.

## Study selection

Criteria for inclusion of a study were (1) study design was a RCT; (2) patients had acute ischemic stroke due to ICA or MCA M1 occlusion; (3) patients had a large ischemic core, defined as ASPECT ≤ 5 on computed tomography (CT) or diffusion-weighted imaging (DWI), or an estimated ischemic-core volume of 50 ml or greater on CT perfusion imaging; (4) trials compared EVT plus medical care versus medical care alone; and (5) functional independence, defined as modified Rankin Scale (mRS) score 0 to 2, at 90 days post-stroke was reported as an endpoint. The mRS is a 7-point ordinal scale (ranging from 0 to 6) commonly used to classify the degree of disability among patients who have experienced a stroke. Studies were excluded if the study design was a registry, case report, case-control, or cohort. We extracted characteristics of each trial, which included patient age, sex, imaging criteria for enrollment, and outcomes assessed. Two investigators (C.-H. L. and M.L.) independently extracted the data and any discrepant judgments were resolved by referencing the original report.

## Study quality assessment

Since all of the included studies were RCTs, we assessed the overall bias (e.g. bias arising from the randomization process, bias due to deviations from intended interventions, bias due to missing outcome data, bias in measurement of the outcome, and bias in selection of the reported result) by using the RoB-2 tool [13].

## Outcomes

The primary outcome was functional independence (mRS score of 0 to 2) at 90 days post-stroke. The secondary outcomes were reduced disability (ordinal shift across the range of mRS scores toward a better outcome, mRS 0–1/2/3/4/5/6) at 90 days post-stroke, ambulation (mRS score of 0 to 3) at 90 days post-stroke, being nondisabled (mRS score of 0 to 1) at 90 days post-stroke, and early neurologic improvement. The safety outcomes were requiring constant care or death (defined as mRS score of 5 to 6) at 90 days post-stroke, death within 90 days post-stroke, and early symptomatic intracranial hemorrhage. We specifically analyzed the outcome of mRS score of 5 to 6 because there is little difference along the disability dimension among mRS score of 5 and 6 outcomes [14] and recent stroke expert consensus reports categorize mRS scores of 5 to 6 together as Very Poor outcome [15].

Subgroup analysis of the primary outcome was conducted based on the imaging modalities used for patient selection for enrollment (non-contrast CT or CT perfusion versus magnetic resonance imaging [MRI]).

## Statistical analysis

The analysis plan was performed on an intention-to-treat basis. We computed the random-effects estimate based on the inverse variance method when 2 or more studies provided sufficient data for a given outcome. Risk ratio (RR) with 95% confidence interval (CI) was used as a measure of the association of EVT plus medical care versus medical care alone with binary outcomes. To aid interpretation of the results of a meta-analysis of risk ratios, we computed an absolute risk reduction or number needed to treat (NNT). In order to do this, an assumed comparator risk (ACR) (risk that the outcome of interest would occur with the comparator intervention) was required. The computation proceeds as follows [16]:

$$\text{number fewer per 1000 (ARR)} = 1000 \times \text{ACR} \times (1 - \text{RR})$$

$$\text{NNT} = \left| \frac{1}{\text{ACR} \times (1 - \text{RR})} \right|$$

Common odds ratio with 95% CI was used as a measure of the association of EVT plus medical care versus medical care alone with an ordinal shift across the range of mRS scores toward a better outcome at 90 days post-stroke. All reported $p$ values were two-sided, with significance set at < 0.05. Given the small number of trials in our study, $I^2$ should be interpreted with caution [17]. Therefore, we assessed heterogeneity primarily through visual inspection of forest plots for each outcome.

The min-max-mean algorithmic method was used to derive the NNT for the ordinal outcome, conducted according to the previously described methodology [18]. The maximum possible NNT compatible with the data was derived by completing the joint outcome table following the rule that every patient who benefits from therapy does so by improving only the minimum possible number of levels compatible with the final trial group outcome distributions. The minimum possible NNT compatible with the data was derived by completing the table following the rule that every patient who benefits from therapy does so by improving the maximum possible number of grades compatible with the group data. The central value within the possible range was obtained by calculating the geometric mean of the minimum and maximum NNTs.

To identify any trial that might have exerted a disproportionate influence on the summary treatment effect, we conducted a sensitivity test by removing each individual trial from the meta-analysis one at a time [11]. Potential small study effects were assessed visually by a funnel plot displaying standard error as the measure of sample size and risk ratio as the measure of treatment effect [11]. The Cochrane Collaboration's Review Manager Software Package (RevMan, version 5.4) was used for this meta-analysis. Grading of Recommendations Assessment, Development and Evaluation (GRADE) was used to assess the certainty of evidence for each outcome, considering factors, such as risk of bias, inconsistency, indirectness, imprecision, and publication bias [11,19].

## Results

We identified 11 full articles for detailed assessment, of which five did not meet the inclusion criteria (S1 Table); therefore, the final analysis included six RCTs (Fig 1) [5–10]. The characteristics of the included trials are shown in Table 1 [5–10]. Overall, 1,870 patients (826 females [44.2%]) with ischemic stroke and a larger moderate or large core due to ICA or MCA M1 occlusion. Five trials [5,6,8–10] enrolled patients with mRS 0–1 before stroke while one trial [7] enrolled patients with mRS 0 to 2 before stroke. Among enrolled patients, 940 were randomly assigned to EVT plus medical care and 930 were randomly assigned to medical care alone. The Recovery by Endovascular Salvage for Cerebral Ultra-Acute Embolism–Japan Large Ischemic Core Trial (RESCUE-Japan LMIT) [5], the Endovascular Therapy in Acute Anterior Circulation Large Vessel Occlusive Patients with a Large Infarct Core (ANGEL-ASPECT) trial [6], the RCT to optimize patient's Selection for Endovascular Treatment in Acute Ischemic Stroke (SLELCT2) [9], and the Thrombectomy for Emergent Salvage of Large Anterior Circulation Ischemic Stroke (TESLA) trial [10] enrolled patients up to 24 hours, whereas the Efficacy and Safety of Thrombectomy in Stroke With Extended Lesion and Extended Time Window (TENSION) trial [7] enrolled patients within 12 hours and the Large Stroke Therapy Evaluation (LASTE) trial [8] enrolled patients within 6.5 hours of last known well. Median baseline NIHSS score ranged from 16 to 22. Median ASPECTS value was 2 in the LASTE trial [8] and 3 or 4 among other included trials. The TESLA trial [10] used ASPECTS value of 2 to 5 on non-contrast CT to select patients while other trials also used advanced imaging modalities, such as MRI or perfusion imaging, to select patients. The RESCUE-Japan LIMIT [5] used mRS value of 0 to 3 as the primary outcome, the ANGEL-ASPECT trial [6], the SELECT2 [9], the TENSION trial [7], and the LASTE trial [8] used an ordinal shift in the distribution of scores on the mRS as the primary outcome, and the TESLA trial [10] used score on utility-weighted mRS as the primary outcome. Successful reperfusion, defined as a Thrombolysis in Cerebral Infarction (TICI) score of 2b or 3, was achieved in 107 of 146 patients (73%) in the TESLA trial's EVT group [10]. This rate was lower compared to other EVT trials, which reported successful reperfusion rates of 80% or higher [5–8]. The RoB-2 for the included trials is summarized in S2 Table. All included trials were judged to have a high risk of bias due to deviations from intended interventions, primarily because they were not blinded.

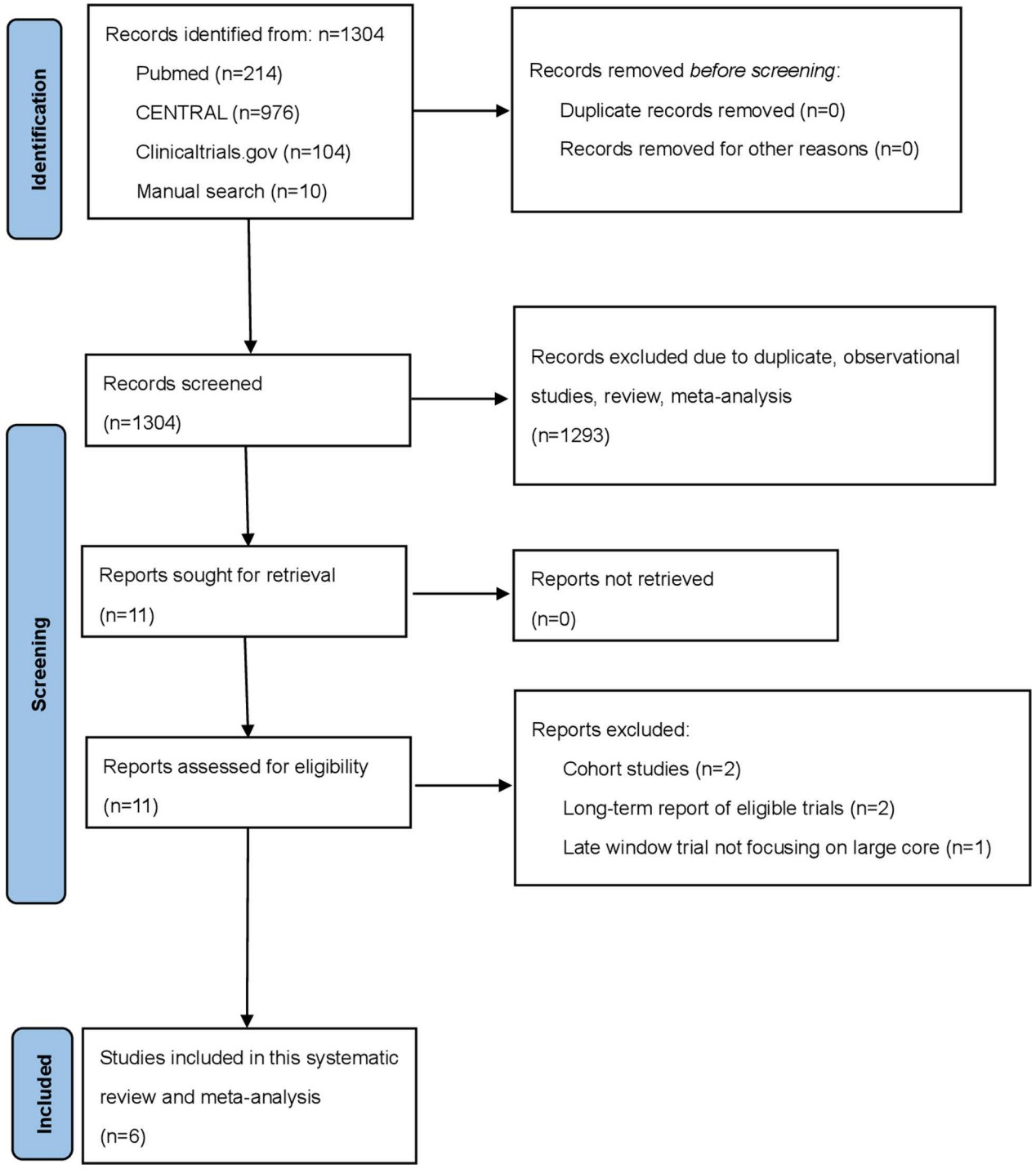

**Fig 1. Flow diagram of study selection.** Flow diagram of study selection.

**Table 1. Baseline characteristics of included trials.**

| | RESCUE-Japan LIMIT [5] | ANGEL-ASPECT [6] | SELECT2 [9] | TESLA [10] | TENSION [7] | LASTE [8] |
|---|---|---|---|---|---|---|
| Countries | Japan | China | USA, Canada, Europe, Australia, New Zealand | USA | Europe, Canada | France |
| Inclusion criteria | Age ≥ 18y, NIHSS ≥ 6, within 24h, occlusion of ICA or M1, ASPECT 3–5 on DWI or CT, pre-stroke mRS 0–1 | Age 18-80y, NIHSS 6–30, within 24h, occlusion of ICA or M1, ASPECT 3–5 on CT (no limit with infarct-core) or ASPECT 0–2 (infarct-core 70–100 ml) or ASPECT>5 (infarct-core 70–100 ml) between 6–24hr, pre-stroke mRS 0–1 | Age 18-85y, NIHSS ≥ 6, within 24h, occlusion of ICA or M1, ASPECT 3–5 on CT or core ≥ 50 ml on CT perfusion, pre-stroke mRS 0–1 | Age 18-85y, NIHSS ≥ 6, within 24h, occlusion of ICA or M1, ASPECT 2–5 on NCCT, pre-stroke mRS 0–1 | Age ≥ 18y, NIHSS ≤ 26, within 12h, occlusion of ICA or M1, ASPECT 3–5 on DWI or CT, pre-stroke mRS 0–2 | Age > 18y, NIHSS ≥ 6, within 6.5h, occlusion of ICA or M1, ASPECT 0–5 on CT or MRI, pre-stroke mRS 0–1 |
| Sample size (% of female), EVT plus medical care vs. medical care alone | 101 (46) vs. 102 (43) | 230 (41) vs. 225 (36) | 178 (40) vs. 174 (43) | 75 (49) vs. 63 (43) | 125 (45) vs. 128 (52) | 159 (48) vs. 165 (47) |
| Age, years, EVT plus medical care vs. medical care alone, *: mean±SD or median (IQR) | 77 ± 10 vs. 76 ± 10 | 68 (61, 73) vs. 67 (59, 73) | 66 (58, 75) vs. 67 (58, 75) | 66 (54, 75) vs. 68 (59, 75) | 73 (62, 81) vs. 74 (64, 80) | 73 (66, 79) vs. 74 (65, 80) |
| Median NIHSS (IQR), EVT plus medical care vs. medical care alone | 22 (18, 26) vs. 22 (17, 26) | 16 (13, 20) vs. 15 (12, 19) | 19 (15, 23) vs. 19 (15, 22) | 19 (15, 23) vs. 18 (15, 21) | 19 (16, 22) vs. 18 (15, 22) | 21 (18, 24) vs. 21 (18, 24) |
| Occlusion site, n **(%),** EVT plus medical care vs. medical care alone | | | | | | |
| ICA | 47 (47) vs. 49 (48) | 83 (36) vs. 81 (36) | 80 (45) vs. 66 (38) | 37 (24) vs. 31 (21) | 41 (33) vs. 37 (29) | 69 (43) vs. 74 (45) |
| MCA M1 | 74 (73) vs. 70 (69) | 145 (63) vs. 142 (63) | 91 (51) vs. 100 (58) | 119 (78) vs. 124 (84) | 83 (66) vs. 88(69) | 88 (55) vs. 91 (55) |
| ASPECTS value, median (IQR), EVT plus medical care vs. medical care alone | 3 (3, 4) vs. 4 (3, 4) | 3 (3, 4) vs. 3 (3, 4) | 4 (3, 5) vs. 4 (4, 5) | 4 vs. 4 (IQR not provided) | 4 vs. 4 (IQR not provided) | 2 (1, 3) vs. 2 (1, 3) |
| Infarct volume, ml, median (IQR), EVT plus medical care vs. medical care alone | 94 (66, 152) vs. 110 (74, 140) | 61 (29, 86) vs. 63 (31, 86) | 82 (57, 118) vs. 79 (62, 111) | NA | NA | 132 (104, 185) vs. 137 (106, 187) |
| IVT, n (%), EVT plus medical care vs. medical care alone | 27 (27) vs. 29 (28) | 66 (29) vs. 63 (28) | 37 (21) vs. 30 (17) | 31 (20) vs. 30 (20) | 49 (39) vs. 44 (34) | 55 (35) vs. 58 (35) |

*(Continued)*

**Table 1.** (Continued)

|  | RESCUE-Japan LIMIT [5] | ANGEL-ASPECT [6] | SELECT2 [9] | TESLA [10] | TENSION [7] | LASTE [8] |
|---|---|---|---|---|---|---|
| Interval between onset and randomization, min, median (IQR), EVT plus medical care vs. medical care alone | 229 (144, 459) vs. 214 (142, 378) | 453 (299, 712) vs. 463 (305, 781) | 587 (349, 919) vs. 544 (316, 920) | 653 (333, 942) vs. 754 (337, 1,027) | 120 (72, 210) vs. 126 (72, 216) | 271 (199, 351) vs. 268 (207, 336) |

**Legends**: Baseline characteristics of included trials

Acronym of trials: RESCUE Japan LIMIT, Recovery by Endovascular Salvage for Cerebral Ultra-Acute Embolism–Japan Large Ischemic Core Trial; ANGEL ASPECT, Endovascular Therapy in Acute Anterior Circulation Large Vessel Occlusive Patients with a Large Infarct Core; SELECT2, Randomized Controlled Trial to Optimize Patient's Selection for Endovascular Treatment in Acute Ischemic Stroke; TESLA, Thrombectomy for Emergent Salvage of Large Anterior Circulation Ischemic Stroke; TENSION, Efficacy and Safety of Thrombectomy in Stroke With Extended Lesion and Extended Time Window; LASTE, Large Stroke Therapy Evaluation.

AF, atrial fibrillation; ASPECTS, Alberta Stroke Program Early Computed Tomography Score; CT, Computed Tomography; DWI, diffuse- weighted imaging; EVT, endovascular thrombectomy; ICA, internal carotid artery; IQR, interquartile range; IVT, intravenous thrombolysis; MCA, middle cerebral artery; M1, M1 segment of middle cerebral artery; mRS, modified Rankin Scale; NA, not available; NCCT, noncontrast computed tomography; NIHSS, National Institutes of Health Stroke Scale; SD: standard deviation; sICH, symptomatic intracranial hemorrhage; TICI, Thrombolysis in Cerebral Infarction; vs., versus

*RESCUE-Japan LIMIT used mean age while other trials used median age.

## Primary outcome

For the primary outcome of functional independence (mRS score of 0 to 2) at 90 days post-stroke, pooled results from the random-effects model showed that EVT plus medical care, compared with medical care alone, was associated with a higher proportion of patients achieving functional independence in acute stroke with a large ischemic core due to ICA or MCA M1 occlusion (six trials; absolute risk, 19.5% versus 7.4%; RR 2.53, 95% CI [1.95, 3.29]; $p < 0.001$; NNT, 9, 95% CI [6,15]) (Fig 2) [5–10]. This represented 113 more patients achieving functional independence per 1,000 treated with EVT plus medical care compared with medical care alone. Heterogeneity might exist among studies, particularly between the TENSION and the TESLA trials, based on visual inspection of the forest plots.

## Secondary outcomes

The distribution of mRS scores in the EVT plus medical care and medical care alone groups is shown in Fig 3. Across these trials, allocation to EVT plus medical care, compared with medical care alone, was associated with a higher proportion of patients with reduced disability at 90 days post-stroke in acute stroke with a large ischemic core due to ICA or MCA M1 occlusion. This included a directionally favorable effect at each of the five health state transitions across the mRS 0–1/2/3/4/5/6 range.

Pooled results showed that EVT plus medical care, compared with medical care alone, was associated with reduced disability at 90 days post-stroke (6 trials; common odds ratio 1.63, 95% CI [1.38, 1.93]; $p < 0.001$; NNT, 4 [minimum possible NNT, 2; maximum possible NNT, 6]) (Fig 4) [5–10]. This represented 270 more patients with reduced disability per 1,000 treated with EVT plus medical care compared with medical care alone. Heterogeneity might exist between the RESCUE-Japan LIMIT and TENSION trials versus the other four trials, based on visual inspection of the forest plots.

For the secondary outcome of ambulation (mRS score of 0 to 3) at 90 days post-stroke, pooled results showed that EVT plus medical care, compared with medical care alone, was associated with a higher proportion of patients achieving ambulation in acute stroke with a large ischemic core due to ICA or MCA M1 occlusion (6 trials; absolute risk, 36.5% versus 19.9%; RR 1.92, 95% CI [1.50, 2.44]; $p < 0.001$; NNT, 6, 95% CI [4,11]) (Fig 5) [5–10]. This represented 183 more

PLOS Medicine

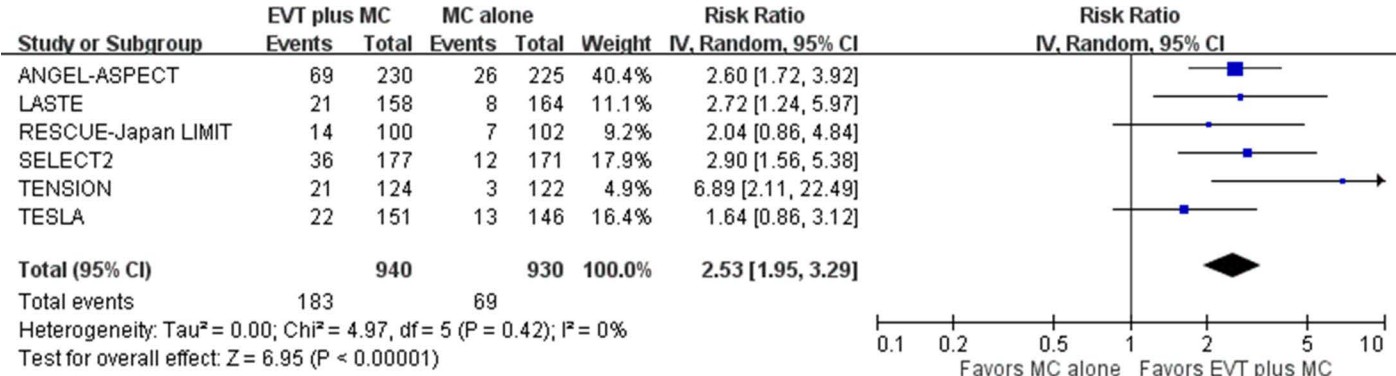

**Fig 2. Functional independence at 90 days post-stroke.** Risk ratio with 95% confidence interval for functional independence, defined as a modified Rankin Scale (mRS) score of 0 to 2, comparing EVT plus medical care with medical care alone in acute stroke with a large ischemic core due to ICA or MCA M1 occlusion at 90 days post-stroke. CI, confidence interval; EVT, endovascular thrombectomy; ICA, internal carotid artery; IV, inverse variance; MC, medical care; MCA, middle cerebral artery; M1, M1 segment of middle cerebral artery.

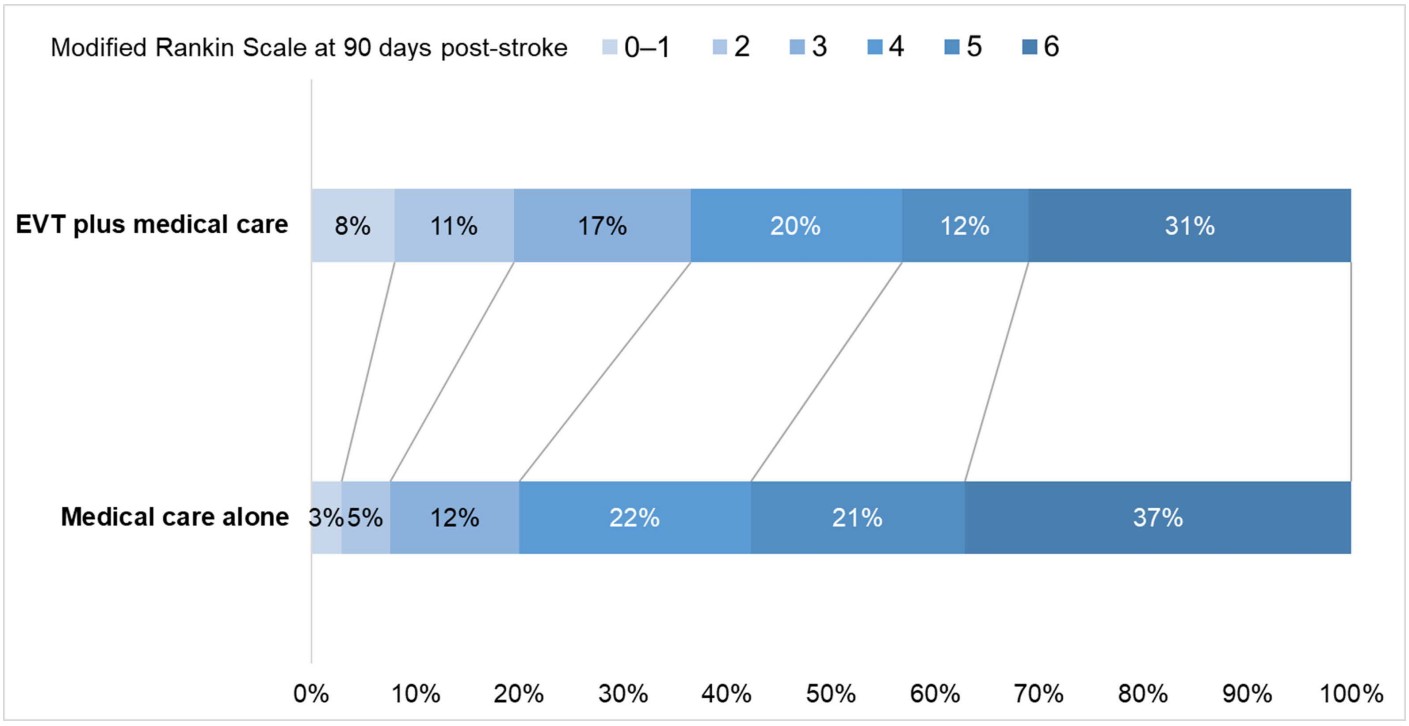

**Fig 3. Distribution of modified Rankin Scale scores at 90 days post-stroke.** A modified Rankin Scale (mRS) score of 0 to 1 indicates nondisabled, 2 indicates disabled but independent, 3 indicates dependent but ambulatory, 4 indicates not ambulatory but not requiring constant care, 5 indicates requiring constant care, and 6 indicates death. EVT, endovascular thrombectomy.

patients achieving ambulation per 1,000 treated with EVT plus medical care compared with medical care alone. Heterogeneity might exist between the ANGEL-ASPECT and TESLA trials versus the other four trials, based on visual inspection of the forest plots.

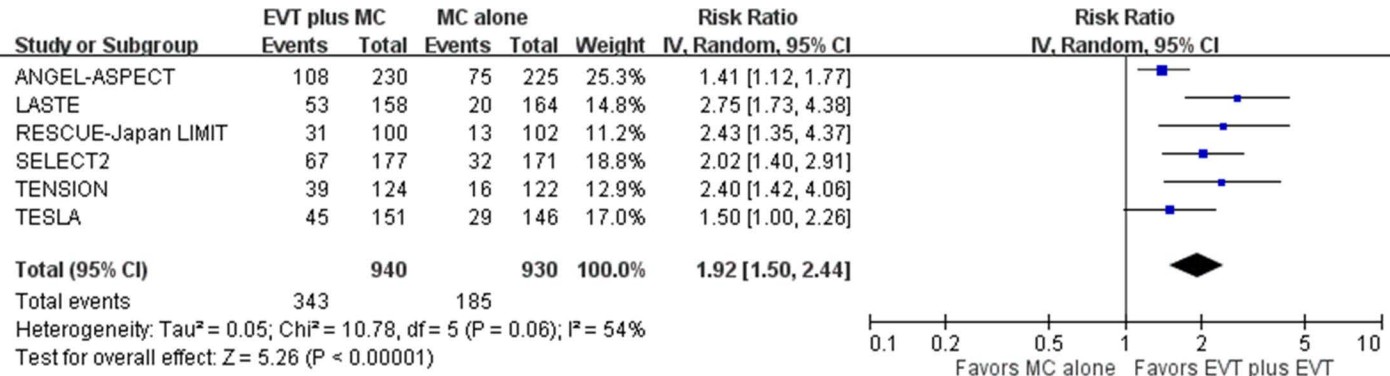

**Fig 4. Reduced disability (ordinal shift of mRS) at 90 days post-stroke.** Odds ratio with a 95% confidence interval for reduced disability, defined as a shift in the distribution of modified Rankin Scale (mRS) scores toward better outcomes in favor of EVT plus medical care compared with medical care alone in acute stroke with a large ischemic core due to ICA or MCA M1 occlusion at 90 days post-stroke. CI, confidence interval; EVT, endovascular thrombectomy; ICA, internal carotid artery; IV, inverse variance; MC, medical care; MCA, middle cerebral artery; M1, M1 segment of middle cerebral artery.

**Fig 5. Ambulation at 90 days post-stroke.** Risk ratio with a 95% confidence interval for achieving ambulation, defined as a modified Rankin Scale (mRS) score of 0 to 3, comparing EVT plus medical care with medical care alone in acute stroke with a large ischemic core due to ICA or MCA M1 occlusion at 90 days post-stroke. CI, confidence interval; EVT, endovascular thrombectomy; ICA, internal carotid artery; IV, inverse variance; MC, medical care; MCA, middle cerebral artery; M1, M1 segment of middle cerebral artery.

For the secondary outcome of being nondisabled (mRS score of 0–1) at 90 days post-stroke, pooled results showed that EVT plus medical care, compared with medical care alone, was associated with a higher proportion of patients achieving nondisabled status in acute stroke with a large ischemic core due to ICA or MCA M1 occlusion (6 trials; absolute risk, 8.0% versus 2.9%; RR 2.64, 95% CI [1.71, 4.08]; $p < 0.001$; NNT, 22, [95% CI [12,49]) (Fig 6) [5–10]. This represented 47 more patients achieving nondisabled status per 1,000 treated with EVT plus medical care compared with medical care alone. Heterogeneity might exist among studies based on visual inspection of the forest plots.

For the secondary outcome of early neurologic improvement, the definition varied across the included trials. Early neurologic improvement was defined differently across the included trials. It was defined as a reduction of at least 8 points in the NIHSS score at 48 hours in the RESCUE-Japan LIMIT [5], at 24 hours in the SELECT2 [9], and as either an NIHSS score of 0 to 2 or a reduction of at least 8 points at 6 days or discharge in the TESLA trial [10]. In the ANGEL-ASPECT trial [6], early neurologic improvement was defined as a reduction of at least 10 points in the NIHSS score or an NIHSS

**Fig 6. Being nondisabled at 90 days post-stroke.** Risk ratio with a 95% confidence interval for achieving nondisabled status, defined as a modified Rankin Scale (mRS) score of 0 to 1, comparing EVT plus medical care with medical care alone in acute stroke with a large ischemic core due to ICA or MCA M1 occlusion at 90 days post-stroke. CI, confidence interval; EVT, endovascular thrombectomy; ICA, internal carotid artery; IV, inverse variance; MC, medical care; MCA, middle cerebral artery; M1, M1 segment of middle cerebral artery.

score of 0 or 1 at 36 hours. In the LASTE trial [8], it was defined as a reduction of at least 10 points in the NIHSS score from the time of presentation to a thrombectomy-capable center to either day 7 or discharge.

Pooled results of the five included trials that reported this outcome showed that EVT plus medical care, compared with medical care alone, was associated with a higher proportion of patients with early neurologic improvement in acute stroke with a large ischemic core due to ICA or MCA M1 occlusion (five trials; absolute risk, 18.6% versus 7.8%; RR 2.35, 95% CI [1.79, 3.10]; $p < 0.001$; NNT, 10, 95% CI [7,17]) (Fig 7) [5,6,8–10]. This represented 106 more patients with early neurologic improvement per 1,000 treated with EVT plus medical care compared with medical care alone. Heterogeneity might exist among studies based on visual inspection of the forest plots.

## Safety outcomes

For the first safety outcome of requiring constant care or death at 90 days post-stroke (mRS score of 5 to 6), pooled results showed that EVT plus medical care, compared with medical care alone, was associated with a lower proportion of patients in this category in acute stroke with a large ischemic core due to ICA or MCA M1 occlusion (six trials; absolute risk, 43.3% versus 57.8%; RR 0.74, 95% CI [0.66, 0.84]; $p < 0.001$; NNT, 7, 95% CI [6,11]) (Fig 8) [5–10]. This represented 150 fewer patients in the mRS 5–6 category per 1000 treated with EVT plus medical care compared with medical care alone. Heterogeneity might exist among studies, particularly between the RESCUE-Japan LIMIT and TESLA trials, based on visual inspection of the forest plots.

For the second safety outcome of death within 90 days post-stroke, pooled results showed that EVT plus medical care, compared with medical care alone, was not associated with a significantly reduced risk of death in acute stroke with a large ischemic core due to ICA or MCA M1 occlusion (six trials; absolute risk, 31.5% versus 36.8%; RR 0.86, 95% CI [0.72, 1.02]; $p = 0.08$) (Fig 9) [5–10]. Heterogeneity might exist among studies, particularly between the ANGEL-ASPECT and TESLA trials versus the LASTE trial, based on visual inspection of the forest plots.

For the third safety outcome, early symptomatic intracranial hemorrhage, the definition varied across the included trials. Early symptomatic intracranial hemorrhage was defined using either the Safe Implementation of Thrombolysis in Stroke Monitoring Study (SITS-MOST) criteria [20] in four trials [5,8–10] or the Heidelberg bleeding classification [21] in three trials [6–8]. The LASTE trial reported early symptomatic intracranial hemorrhage using both the SITS-MOST criteria and the Heidelberg bleeding classification [8]. For our pooled analysis, we used outcomes based on the SITS-MOST criteria.

**Fig 7. Early neurologic improvement.** Risk ratio with a 95% confidence interval for early neurologic improvement, comparing EVT plus medical care with medical care alone in acute stroke with a large ischemic core due to ICA or MCA M1 occlusion. CI, confidence interval; EVT, endovascular thrombectomy; ICA, internal carotid artery; IV, inverse variance; MC, medical care; MCA, middle cerebral artery; M1, M1 segment of middle cerebral artery.

**Fig 8. Requiring constant care or death at 90 days post-stroke.** Risk ratio with a 95% confidence interval for requiring constant care or death, defined as a modified Rankin Scale (mRS) score of 5 to 6, comparing EVT plus medical care with medical care alone in acute stroke with a large ischemic core due to ICA or MCA M1 occlusion at 90 days post-stroke. CI, confidence interval; EVT, endovascular thrombectomy; ICA, internal carotid artery; IV, inverse variance; MC, medical care; MCA, middle cerebral artery; M1, M1 segment of middle cerebral artery.

Pooled results showed that EVT plus medical care, compared with medical care alone, was associated with a non-significantly higher proportion of patients with early symptomatic intracranial hemorrhage in acute stroke with a large ischemic core due to ICA or MCA M1 occlusion (six trials; absolute risk, 4.4% versus 2.7%; RR 1.65, 95% CI [1.00, 2.70]; $p = 0.05$) (Fig 10) [5–10]. Heterogeneity might exist between the SELECT2 and the other five trials, based on visual inspection of the forest plots.

The funnel plot for the primary outcome did not show major asymmetry (S1 Fig).

Assessment of the quality of the evidence using the GRADE approach is presented in Table 2. The GRADE assessment indicated high-certainty evidence for the outcomes of functional independence, being nondisabled, and early neurologic improvement, and moderate-certainty evidence for reduced disability, ambulation, requiring constant care or death, death, and early symptomatic intracranial hemorrhage.

Subgroup analysis showed that the benefit of EVT in achieving functional independence was not significantly different between patients selected by non-contrast CT or CTP versus those selected by MRI, based on visual inspection of the forest plots (RR 2.62, 95% CI [1.76, 3.91] versus RR 2.39, 95% CI [1.34, 4.27]) (S2 Fig).

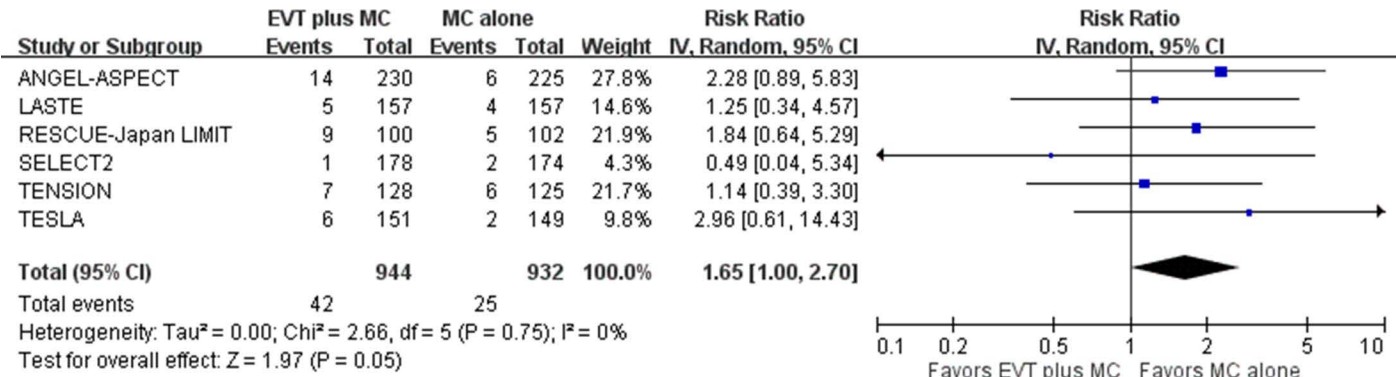

**Fig 9. Death within 90 days post-stroke.** Risk ratio with a 95% confidence interval for death, comparing EVT plus medical care with medical care alone in acute stroke with a large ischemic core due to ICA or MCA M1 occlusion within 90 days post-stroke. CI, confidence interval; EVT, endovascular thrombectomy; ICA, internal carotid artery; IV, inverse variance; MC, medical care; MCA, middle cerebral artery; M1, M1 segment of middle cerebral artery.

**Fig 10. Early symptomatic intracranial hemorrhage.** Risk ratio with a 95% confidence interval for early symptomatic intracranial hemorrhage, comparing EVT plus medical care with medical care alone in acute stroke with a large ischemic core due to ICA or MCA M1 occlusion. CI, confidence interval; EVT, endovascular thrombectomy; ICA, internal carotid artery; IV, inverse variance; MC, medical care; MCA, middle cerebral artery; M1, M1 segment of middle cerebral artery.

Sensitivity test excluding individual trials serially yielded pooled results similar to the overall pooled estimates of the primary outcome, indicating that no single study had a disproportionate influence on the overall effect size (S3–S8 Figs). Sensitivity tests excluding individual trials serially for primary, secondary, and safety outcomes are presented in Tables 3 and 4.

## Discussion

Our systematic review and meta-analysis, which included data from six RCTs involving 1,870 patients with acute stroke and a large core due to ICA or MCA M1 occlusion who were nondisabled before stroke, demonstrated that compared to medical care alone, EVT plus medical care was associated with clinical benefit. Specifically, EVT plus medical care was linked to better functional outcomes, including increased rates of functional independence, reduced disability, improved ambulation, and early neurological improvement. For every 1,000 patients, EVT led to 270 patients with reduced disability, including 113 more patients with functional independence than medical care. Additionally, this approach reduced the risk of requiring constant care or death at 90 days post-stroke.

**Table 2. Grading of Recommendations Assessment, Development and Evaluation (GRADE) of outcomes.**

| Quality assessment | | | | | | | Summary of findings | | | | Quality |
|---|---|---|---|---|---|---|---|---|---|---|---|
| | | | | | | | Event, No./Total, No. | | Effect | | |
| Outcomes, No. of studies | Design | Study limitation | Inconsistency | Indirectness | Imprecision | Publication bias | EVT plus medical care | Medical care alone | Relative (95% CI) | Absolute | |
| Functional independence (mRS 0–2), n=6 | RCT | Lack of blinding | Inconsistency not important | No indirectness | No serious imprecision | Publication bias undetected | 183/940 | 69/930 | 2.53 (1.95, 3.29) | 113 more per 1,000 (70, 169) | High |
| Reduced disability (shift of mRS), n=6 | RCT | Lack of blinding | Moderate inconsistency | No indirectness | No serious imprecision | Publication bias undetected | NA | NA | Odds ratio 1.63 (1.38, 1.93) | 270 more per 1,000 (166, 555) | Moderate |
| Ambulation (mRS 0–3), n=6 | RCT | Lack of blinding | Substantial inconsistency | No indirectness | No serious imprecision | Publication bias undetected | 343/940 | 185/930 | 1.92 (1.50, 2.44) | 183 more per 1,000 (99, 286) | Moderate |
| Being nondisabled (mRS 0–1), n=6 | RCT | Lack of blinding | Inconsistency not important | No indirectness | No serious imprecision | Publication bias undetected | 75/940 | 27/930 | 2.64 (1.71, 4.08) | 47 more per 1,000 (20, 89) | High |
| Early neurologic improvement, n=5 | RCT | Lack of blinding | Inconsistency not important | No indirectness | No serious imprecision | Publication bias undetected | 150/808 | 63/804 | 2.35 (1.79, 3.10) | 106 more per 1,000 (62, 165) | High |
| Requires constant care or death (mRS 5–6), n=6 | RCT | Lack of blinding | Moderate inconsistency | No indirectness | No serious imprecision | Small degree of publication bias detected | 407/940 | 538/930 | 0.74 (0.66, 0.84) | 150 fewer per 1,000 (92, 196) | Moderate |
| Death, n=6 | RCT | Lack of blinding | Moderate inconsistency | No indirectness | Some imprecision | Publication bias undetected | 295/937 | 343/932 | 0.86 (0.72, 1.02) | NA | Moderate |
| Early symptomatic intracranial hemorrhage n=6 | RCT | Lack of blinding | Inconsistency not important | No indirectness | Some imprecision | Small degree of publication bias detected / Small degree of publication bias detected | 42/944 | 25/932 | 1.65 (1.00, 2.70) | NA | Moderate |

CI, confidence interval; EVT, endovascular thrombecomy; mRS, modified Rankin Scale; NA, not applicable; No, number; RCT, randomized controlled trial.

**Table 3. Sensitivity tests for primary and secondary outcomes.**

| | Functional independence, RR (95% CI), 6 trials | Reduced disability, OR (95% CI), 6 trials | Ambulation, RR (95% CI), 6 trials | Being nondisabled, RR (95% CI), 6 trials | Early neurologic improvement, RR (95% CI), 5 trials |
|---|---|---|---|---|---|
| All trials pooled | 2.53 (1.95, 3.29) | 1.63 (1.38, 1.93) | 1.92 (1.50, 2.44) | 2.64 (1.71, 4.08) | 2.35 (1.79, 3.10) |
| ANGEL-ASPECT excluded | 2.52 (1.72, 3.71) | 1.73 (1.42, 2.11) | 2.10 (1.69, 2.59) | 2.33 (1.37, 3.96) | 2.31(1.68, 3.16) |
| LASTE excluded | 2.52 (1.82, 3.48) | 1.66 (1.33, 2.08) | 1.77 (1.41, 2.23) | 2.67 (1.68, 4.23) | 2.24 (1.55, 3.23) |
| RESCUE-Japan LIMIT excluded | 2.60 (1.91, 3.55) | 1.56 (1.33, 1.82) | 1.86 (1.43, 2.42) | 2.77 (1.75, 4.38) | 2.18 (1.62, 2.94) |
| SELECT2 excluded | 2.46 (1.77, 3.43) | 1.70 (1.36, 2.13) | 2.02 (1.40, 2.91) | 2.54 (1.60, 4.04) | 2.57 (1.90, 3.48) |
| TENSION excluded | 2.41 (1.84, 3.15) | 1.53 (1.34, 1.75) | 1.86 (1.43, 2.42) | 2.49 (1.58, 3.93) | NA |
| TESLA excluded | 2.76 (2.07, 3.67) | 1.68 (1.38, 2.04) | 2.04 (1.53, 2.72) | 3.11 (1.87, 5.18) | 2.53 (1.77, 3.60) |

Sensitivity tests excluding individual trials serially for primary and secondary outcomes, comparing EVT plus medical care with medical care alone in acute stroke with a large ischemic core due to ICA or MCA M1 occlusion.

CI, confidence interval; EVT, endovascular thrombectomy; ICA, internal carotid artery; MCA, middle cerebral artery; M1, M1 segment of middle cerebral artery; NA, not applicable; OR, odds ratio; RR, risk ratio.

The magnitude of benefit of EVT for patients with acute stroke and a large ischemic core due to ICA or MCA M1 occlusion in this study was less than that previously demonstrated for patients with a small to moderate ischemic core. Among 1,000 treated patients, EVT led to 270 with reduced disability in the large core group, compared with 380 in the small to moderate core group [1]. The locus of benefit along the disability spectrum also differed, with benefit for nondisabled (mRS 0–1) and independent (mRS 0–2) outcomes greater for patients with a small to moderate ischemic core but benefit for ambulatory (mRS 0–3) and self-care capable (mRS 0–4) outcomes similar across core sizes. For example, for patients with a small to moderate versus a large ischemic core, the increase in mRS 0–2 outcomes per 1,000 patients was 195 versus 113, while for mRS 0–3 outcomes the increase was 200 versus 183 [1]. This diminished benefit in extensive ischemic cores, especially for the highest functional outcomes, likely stems from patients having already sustained a large burden of irreversible brain injury that cannot be salvaged despite successful reperfusion with EVT. The findings of this study provide support for the ethics of pursuing EVT in patients with acute stroke and a large ischemic core due to ICA or MCA M1 occlusion. The magnitude of benefit on the primary functional independence (mRS 0–2) outcome was substantial—an absolute

**Table 4. Sensitivity tests for safety outcomes.**

| | Requiring constant care or death, RR (95% CI), 6 trials | Death, RR (95% CI), 6 trials | Early symptomatic intracranial hemorrhage, RR (95% CI), 6 trials |
|---|---|---|---|
| All trials pooled | 0.74 (0.66, 0.84) | 0.86 (0.72, 1.02) | 1.65 (1.00, 2.70) |
| ANGEL-ASPECT excluded | 0.73 (0.64, 0.83) | 0.82 (0.69, 0.99) | 1.45 (0.81, 2.60) |
| LASTE excluded | 0.75 (0.65, 0.87) | 0.92 (0.80, 1.06) | 1.73 (1.01, 2.95) |
| RESCUE-Japan LIMIT excluded | 0.77 (0.69, 0.85) | 0.87 (0.72, 1.05) | 1.60 (0.91, 2.79) |
| SELECT2 excluded | 0.73 (0.64, 0.85) | 0.84 (0.68, 1.04) | 1.74 (1.05, 2.88) |
| TENSION excluded | 0.76 (0.67, 0.87) | 0.88 (0.71, 1.09) | 1.82 (1.04, 3.19) |
| TESLA excluded | 0.72 (0.64, 0.80) | 0.82 (0.69, 0.98) | 1.54 (0.92, 2.60) |

Sensitivity tests excluding individual trials serially for safety outcomes, comparing EVT plus medical care with medical care alone in acute stroke with a large ischemic core due to ICA or MCA M1 occlusion.

CI, confidence interval; EVT, endovascular thrombectomy; ICA, internal carotid artery; MCA, middle cerebral artery; M1, M1 segment of middle cerebral artery; RR, risk ratio.

increase of 12%, yielding a NNT of 9. While not as large as the extraordinary benefit of EVT for patients with a small ischemic core, this effect magnitude on functional independence actually exceeds that of thrombolytic therapy administered within 3 hours of onset (absolute increase 8.5%, NNT, 11) [22]. It also well exceeds the 3.1%–5% absolute increase judged by neurologists and neurointerventionalists as the minimal clinically important difference for adopting into clinical practice a new population for endovascular thrombectomy [23]. Given the significant clinical benefit demonstrated, allocating resources for EVT in large-core stroke patients appears to be evidence-based and clinically justified.

In patients randomized to the EVT plus medical care group in the SELECT2, clinical outcomes worsened as estimates of ischemic injury increased [24]. This study-level meta-analysis could not define the lower boundary of ASPECTS value for a clinical benefit of EVT. A meta-analysis of individual patient-level data from acute stroke with small or moderate ischemic core due to ICA or MCA M1 occlusion did not find any imaging biomarker that could be used as a treatment effect modifier to decide which patients should be treated with EVT or not [25]. Furthermore, since preservation of the high cortical regions was more strongly associated with improved outcomes compared to the deep regions [26], not all low ASPECTS infarcts are highly disabling infarcts. Also, initial CT perfusion in the early hours usually overestimates the final infarct core [27]. Relying on the CT perfusion mismatch concept to select patients for EVT will inadvertently exclude individuals who could still benefit from reperfusion. About 15% to 25% of patients had permanent reversal of DWI after EVT, associated with good clinical outcome, which highlights the pitfalls of DWI to define ischemic core [28,29]. Also, an observational study showed that recanalization was associated with improved functional outcomes in patients without salvage tissue on CT perfusion [30]. A recent meta-analysis of a patient with ASPECT score value of 0 to 2 in the ANGEL-ASPECT trial [6] and the SELECT2 [9] suggest that EVT might still be beneficial for patients with ASPECT score < 3 [31] and similar finding was observed in the post-hoc analysis of the TENSION trial [7]. Still, the RESCUE-Japan LIMIT suggested that EVT was cost-effective for participants with an ASPECTS 4 to 5, but not for those with an ASPECTS value of 3 or less from both the US and Japan perspectives [32,33]. Finally, it is crucial to move beyond just the quantity of infarcted tissue and consider the infarct's location and eloquent areas- the quality and functionality of the non-infarcted brain tissue. A future nuanced perspective that incorporates handedness or better dominant brain side could significantly enhance patient selection and outcomes in EVT. The current paper underscores the complexity of optimizing treatment strategies for patients with acute stroke and a large ischemic core due to ICA or MCA M1 occlusion and highlights areas for future research, such as refining or redefining imaging selection criteria to optimize outcomes for patients with a large ischemic core.

The primary outcome varied among the included trials and the TESLA trial [10] was the only trial that failed to show superiority of EVT to medical care in its original primary outcome. Also, the TESLA trial did not show a significant benefit of EVT plus medical care over medical care alone for improving functional independence, reducing disability, and reducing patients requiring constant care or dead [10]. Successful reperfusion was only 73% in the TESLA trial [10] whereas 80% or more of successful reperfusion was found in other trials [5–8]. Since poor reperfusion after EVT was associated with worse outcomes compared with medical care alone [34], a relatively lower rate of successful reperfusion in the TESLA trial [10] was likely to hamper the functional outcome in patients allocated to the EVT plus medical care group.

A similar meta-analysis was published by Chen et al. [35], which also investigated the efficacy of EVT in patients with acute stroke and a large ischemic core. The current study was distinct from the study by Chen et al. [35] in several aspects. First, Chen et al. published their meta-analysis of six randomized trials in January 2024, predating the publication of the LASTE trial (May 2024) [8] and the online publication of the TESLA trial (September 2024) [10]. Although Chen et al. [35] cited the LASTE trial protocol [36], this protocol contained no outcome data. Therefore, the accuracy of the data from these two trials in Chen's meta-analysis cannot be verified [35]. Second, given that these trials included patients with a large ischemic core, it was essential to determine whether EVT plus medical care reduced the risk of the very poor outcome (mRS 5 to 6) compared with medical care alone. Unlike Chen's analysis [35], this meta-analysis demonstrated that EVT plus medical care significantly reduced the very poor outcome compared with medical care alone.

This study has limitations. First, the level of ASPECTS value below which EVT should not be recommended is not known based on the evidence currently available. Comprehensive cost-effective analyses may be needed before EVT can be applied universally for patients with acute stroke and a large ischemic core due to ICA or MCA M1 occlusion. Second, the imaging modality to define large ischemic cores varied among included trials. The trial that predominantly relied on non-contrast CT scans for identifying large ischemic cores had enrolled patients within 12 hours of stroke symptom onset [7]. Consequently, for selecting appropriate patients with a large ischemic core for EVT in cases where the last known well time falls between 12 and 24 hours, more advanced imaging techniques such as CT perfusion or MRI might be required. Third, per the original definition of large core, greater than one-third of the MCA territory or 100 ml, the current analysis actually addresses patients with a combination of larger moderate core and true large core, rather than patients with large core alone.

In conclusion, this meta-analysis of RCT data suggests that among patients with acute stroke and a larger moderate or large ischemic core due to ICA or MCA M1 occlusion who were nondisabled prior to stroke, EVT plus medical care, compared with medical care alone, may be associated with improved functional independence, reduced disability, and reduced rates of severe disability or death at 90 days post-stroke. Further individual patient-level pooled analysis and cost-effectiveness analysis are warranted to better define the lower ASPECTS threshold and the upper infarct volume limit on CT perfusion or DWI that may best identify patients most likely to benefit from EVT.

## Supporting information

**S1 Checklist. PRISMA checklist.**
(DOCX)

**S1 Table. Studies excluded after detailed assessment.**
(DOCX)

**S2 Table. Risk of bias 2 (RoB-2) of included trials.**
(DOCX)

**S1 Fig. Funnel plot.** Funnel plot of included trials. RR, risk ratio; SE, standard error.
(TIF)

**S2 Fig. Subgroup Analysis of Treatment Effect by Imaging Selection Modality.** Treatment effect of EVT plus medical care compared with medical care alone on functional independence stratified by imaging modality. CT: computed tomography; EVT, endovascular thrombectomy; MRI: magnetic resonance imaging.
(TIF)

**S3 Fig. Sensitivity test excluding ANGEL-ASPECT.** Sensitivity analysis of functional independence at 90 days, comparing EVT plus medical care with medical care alone in acute stroke with a large ischemic core due to ICA or MCA M1 occlusion, after excluding the ANGEL-ASPECT trial. CI, confidence interval; EVT, endovascular thrombectomy; ICA, internal carotid artery; IV, inverse variance; MC, medical care; MCA, middle cerebral artery; M1, M1 segment of middle cerebral artery.
(TIF)

**S4 Fig. Sensitivity test excluding LASTE.** Sensitivity analysis of functional independence at 90 days, comparing EVT plus medical care with medical care alone in acute stroke with a large ischemic core due to ICA or MCA M1 occlusion, after excluding the LASTE trial. CI, confidence interval; EVT, endovascular thrombectomy; ICA, internal carotid artery; IV, inverse variance; MC, medical care; MCA, middle cerebral artery; M1, M1 segment of middle cerebral artery.
(TIF)

**S5 Fig. Sensitivity test excluding RESCUE-Japan LIMIT.** Sensitivity analysis of functional independence at 90 days, comparing EVT plus medical care with medical care alone in acute stroke with a large ischemic core due to ICA or MCA M1 occlusion, after excluding the RESCUE-Japan LIMIT. CI, confidence interval; EVT, endovascular thrombectomy; ICA, internal carotid artery; IV, inverse variance; MC, medical care; MCA, middle cerebral artery; M1, M1 segment of middle cerebral artery.
(TIF)

**S6 Fig. Sensitivity test excluding SELECT2.** Sensitivity analysis of functional independence at 90 days, comparing EVT plus medical care with medical care alone in acute stroke with a large ischemic core due to ICA or MCA M1 occlusion, after excluding the SELECT2 trial. CI, confidence interval; EVT, endovascular thrombectomy; ICA, internal carotid artery; IV, inverse variance; MC, medical care; MCA, middle cerebral artery; M1, M1 segment of middle cerebral artery.
(TIF)

**S7 Fig. Sensitivity test excluding TENSION.** Sensitivity analysis of functional independence at 90 days, comparing EVT plus medical care with medical care alone in acute stroke with a large ischemic core due to ICA or MCA M1 occlusion, after excluding the TENSION trial. CI, confidence interval; EVT, endovascular thrombectomy; ICA, internal carotid artery; IV, inverse variance; MC, medical care; MCA, middle cerebral artery; M1, M1 segment of middle cerebral artery.
(TIF)

**S8 Fig. Sensitivity test excluding TESLA.** Sensitivity analysis of functional independence at 90 days, comparing EVT plus medical care with medical care alone in acute stroke with a large ischemic core due to ICA or MCA M1 occlusion, after excluding the TESLA trial. CI, confidence interval; EVT, endovascular thrombectomy; ICA, internal carotid artery; IV, inverse variance; MC, medical care; MCA, middle cerebral artery; M1, M1 segment of middle cerebral artery.
(TIF)

**S1 Data. Data used for the analysis of dichotomous outcomes at 90 days post-stroke in each trial.**
(XLSX)

**S2 Data. Data used for the analysis of distribution of modified Rankin Scale (mRS) scores at 90 days post-stroke.**
(XLSX)

## Acknowledgment

We thank Dr. Chih-Hao Chen for his assistance in preparing Fig 3.

## Author contributions

**Conceptualization:** Chun-Hsien Lin, Meng Lee, Jeffrey L. Saver.

**Data curation:** Meng Lee, Bruce Ovbiagele, David S. Liebeskind.

**Formal analysis:** Chun-Hsien Lin, Meng Lee, Borja Sanz-Cuesta.

**Funding acquisition:** Meng Lee.

**Investigation:** Meng Lee.

**Methodology:** Meng Lee.

**Resources:** Meng Lee.

**Supervision:** Bruce Ovbiagele, David S. Liebeskind, Jeffrey L. Saver.

**Writing – original draft:** Chun-Hsien Lin.

**Writing – review & editing:** Meng Lee, Bruce Ovbiagele, David S. Liebeskind, Borja Sanz-Cuesta, Jeffrey L. Saver.

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
