## [Editor Report · Decision Letter 0]

4 Oct 2024

Dear Dr Lee,

Thank you for submitting your manuscript entitled "Endovascular Thrombectomy for Acute Stroke with Large Ischemic Core: A Meta-analysis of Randomized Clinical Trials" for consideration by PLOS Medicine.

Your manuscript has now been evaluated by the PLOS Medicine editorial staff as well as by an academic editor with relevant expertise and I am writing to let you know that we would like to send your submission out for external peer review.

Please re-submit your manuscript within two working days, i.e. by Oct 08 2024 11:59PM.

Kind regards,

Philippa C. Dodd, MBBS MRCP PhD

Senior Editor

PLOS Medicine

---

## [Decision Letter · Decision Letter 1]

20 Dec 2024

Dear Dr Lee,

Many thanks for submitting your manuscript "Endovascular Thrombectomy for Acute Stroke with Large Ischemic Core: A Meta-analysis of Randomized Clinical Trials" (PMEDICINE-D-24-03296R1) to PLOS Medicine. The paper has been reviewed by subject experts and a statistician; their comments are included below and can also be accessed here: [LINK]

As you will see, the reviewers found the study to be interesting, but raised several points for clarification. After discussing the paper with the editorial team, I'm pleased to invite you to revise the paper in response to the reviewers' comments. We plan to send the revised paper to some or all of the original reviewers, and we cannot provide any guarantees at this stage regarding publication.

We ask that you submit your revision by Jan 10 2025. However, if this deadline is not feasible, please contact me by email, and we can discuss a suitable alternative.

Don't hesitate to contact me directly with any questions (atosun@plos.org).

Best regards,

Alexandra Tosun, PhD

Associate Editor

[on behalf of]

Philippa Dodd, MBBS MRCP PhD

Senior Editor

PLOS Medicine

Comments from the reviewers:

Reviewer #1: I've been tasked with reviewing a study level meta-analysis of the six major large core trials of EVT vs. medical care. I am reviewing a revised version of the manuscript that has already gone one round of peer review. Overall the study reads well and the methodology of the systematic review and meta-analysis is sound. I have no major criticisms. The limitations highlight the major drawbacks of a study level meta-analysis. Some minor comments:

1. In the discussion, the authors state: "In EVT-treated patients in an individual participant level analysis of a subset of the completed trials, clinical outcomes worsened as ischemic injury estimates increased" - please correct, this was an individual patient level analysis of a single trial (SELECT2)

2. In the discussion, the authors state: " A meta-analysis of individual patient level data did not find any imaging biomarker that could be used as a treatment effect modifier to decide which AIS-LVO patients should be treated with EVT or not" - please note that this IPD meta-analysis was of small/moderate core patients (HERMES).

3. One suggested additional analysis - given the heterogenity in the imaging modality used for each trial, it may be worth performing a sensivity analysis looking at the trials that primarily used Non-contrast CT or CTP (TENSION, TESLA, SELECT2, ANGLES-ASPECT) and the trials that primarily used MRI (LASTE, RESCUE-LIMIT Japan).

Reviewer #2: See attachment

Michael Dewey

Reviewer #3: The authors present a meta-analysis of the 6 randomized controlled trials comparing medical management to endovascular therapy for large vessel occlusion acute ischemic stroke in patients with large ischemic core. Pooled results are reported to show at 90 days, EVT (+MM) vs MM alone was associated with greater functional independence, defined as mRS 0-2 (RR 2.53, NNT 9), and reduced disability (common OR 1.63, NNT 4). EVT had a lower risk of requiring constant care or death (NNT 7). Rates of death and early sICH did not significantly differ between EVT vs MM alone. The authors conclude EVT may be associated with improved functional outcomes and reduced severe disability or death at 90d for acute LVO stroke with large ischemic core.

A meta-analysis evaluating the same 6 trials by Chen et al was already published in JNIS earlier this year. There are some statistics provided here that were not reported in the Chen paper (ex NNT for various outcomes), but the existence of a pre-existing meta-analysis of the exact same trials (and what this article provides in terms of novelty) is not acknowledged/provided. I would recommend doing this.

The authors' conclusion, as stated in the abstract, does not contain any caveats aside from the word "may". In their discussion, the authors acknowledge that the outcomes in the trials analyzed are not equivalent to patients treated with small/moderate sized cores in prior trials (which is not surprising). It also acknowledges some (but not all) of the limitations of the large core trials included in the meta-analysis. However, their conclusion--which is often the only thing many practitioners read—does not acknowledge any of these limitations and may be interpreted by readers only skimming the abstract to imply any patient population would benefit similarly from EVT as those with small/moderate core infarcts. The conclusion is incomplete and merits editing to account for the many limitations the data from these trials provide (Ex--should a 95 year old patient with a baseline mRS 3, an ASPECTS of 2, a dominant M2 occlusion, a LSW 16 hours prior undergo EVT based on the results of this meta-analysis? The conclusion provided implies yes, whereas I and I'd venture to guess most physicians familiar with these trials would argue absolutely not).

While the CI includes 1.0 for early sICH, it certainly trended toward being higher in the EVT group and probably merits a mention of this.

Reviewer #4: This manuscript presents a comprehensive meta-analysis on endovascular therapy (EVT) for large ischemic core strokes, a topic of great clinical importance. The adherence to PRISMA guidelines, and robust statistical methods enhances the reliability of the findings.

The manuscript is well-organized, with clear delineation of methods, results, and discussion. Phrasing "940 been randomly assigned" should be corrected to "940 were randomly assigned."

The authors conclude that EVT may be justified despite diminished benefits in patients with large cores. As this is supported by the data, it may warrant discussion of ethical considerations, such as resource allocation.

Finally, the authors deserve commendation for highlighting the critical issue of initial CT perfusion in the early ours overestimating the final infarct core. Relying on the CT perfusion mismatch concept to select patients for endovascular treatment (EVT) will inadvertently exclude individuals who could still benefit from reperfusion. This is an important factor to consider and may significantly impact analysis. Also, it is crucial to finally move beyond just the quantity of infarcted tissue and consider the infarct's location and eloquent areas- the quality and functionality of the non-infarcted brain tissue. A future nuanced perspective that incorporates handedness or better dominant brain side could significantly enhance patient selection and outcomes in EVT. The paper underscores the complexity of optimizing treatment strategies for stroke patients and highlight areas for future research, such as refining or redefining imaging selection criteria to optimize outcomes for patients with large ischemic cores.

---

* Please upload any figures associated with your paper as individual TIF or EPS files with 300dpi resolution at resubmission; please read our figure guidelines for more information on our requirements: http://journals.plos.org/plosmedicine/s/figures. While revising your submission, please upload your figure files to the PACE digital diagnostic tool, https://pacev2.apexcovantage.com/. PACE helps ensure that figures meet PLOS requirements. To use PACE, you must first register as a user. Then, login and navigate to the UPLOAD tab, where you will find detailed instructions on how to use the tool. If you encounter any issues or have any questions when using PACE, please email us at PLOSMedicine@plos.org.

* FINANCIAL DISCLOSURES: The funding statement should include: specific grant numbers, initials of authors who received each award, URLs to sponsors’ websites. Also, please state whether any sponsors or funders (other than the named authors) played any role in study design, data collection and analysis, the decision to publish, or preparation of the manuscript. If they had no role in the research, include this sentence: “The funders had no role in study design, data collection and analysis, decision to publish, or preparation of the manuscript.”

* COMPETING INTEREST: All authors must declare their relevant competing interests per the PLOS policy, which can be seen here: https://journals.plos.org/plosmedicine/s/competing-interests

For authors with ties to industry, please indicate whether any of the interests has a financial stake in the results of the current study.

FIGURES AND TABLES

SUPPLEMENTARY MATERIAL

REFERENCES

* Where website addresses are cited, please include the complete URL and specify the date of access (e.g. [accessed: 12/06/2024]).

STUDY TYPE-SPECIFIC REQUESTS

* Please report your SR/MA according to the PRISMA guidelines provided at the EQUATOR site. http://www.equator-network.org/reporting-guidelines/prisma/. Please provide the completed PRISMA checklist as Supporting Information. When completing the checklist, please use section and paragraph numbers, rather than page numbers. Please add the following statement, or similar, to the Methods: "This study is reported as per the Preferred Reporting Items for Systematic Reviews and Meta-Analyses (PRISMA) guideline (S1 Checklist)."

* Abstract: Please report your abstract according to PRISMA for abstracts (https://doi.org/10.1371/journal.pmed.1001419) following the PLOS Medicine abstract structure (Background, Methods and Findings, Conclusions). Please ensure you provide dates of search, data sources, number of studies included, types of study designs included, eligibility criteria, and synthesis/appraisal methods.

* Please note that we expect searches to be updated to within 6 months of the time of submission.

---

## [Decision Letter · Decision Letter 2]

17 Mar 2025

Dear Dr. Lee,

Thank you very much for re-submitting your manuscript "Endovascular Thrombectomy for Acute Stroke with Large Ischemic Core: A Systematic Review and Metaanalysis of Randomized Controlled Trials" (PMEDICINE-D-24-03296R2) for review by PLOS Medicine.

Thank you for your patience in waiting for this decision, and for your detailed responses to the reviewers' and editors' comments. I have discussed the paper with my colleagues, and it has also been seen again by the statistical reviewer. The changes made to the paper were satisfactory to the reviewer. As such, we intend to accept the paper for publication, pending your attention to the editors' comments below in a further revision. When submitting your revised paper, please once again include a detailed point-by-point response to the editorial comments.

[LINK]

In revising the manuscript for further consideration here, please ensure you address the specific points made by each reviewer and the editors. In your rebuttal letter you should indicate your response to the reviewers' and editors' comments and the changes you have made in the manuscript. Please submit a clean version of the paper as the main article file. A version with changes marked must also be uploaded as a marked up manuscript file. Please also check the guidelines for revised papers at http://journals.plos.org/plosmedicine/s/revising-your-manuscript for any that apply to your paper.

We ask that you submit your revision within 1 week (Mar 24 2025). However, if this deadline is not feasible, please contact me by email, and we can discuss a suitable alternative.

Please do not hesitate to contact me directly with any questions (atosun@plos.org). If you reply directly to this message, please be sure to 'Reply All' so your message comes directly to my inbox.

We look forward to receiving the revised manuscript. 

Sincerely,

Alexandra Tosun, PhD

Associate Editor 

PLOS Medicine

plosmedicine.org

Comments from Reviewers:

Reviewer #2: The authors have addressed my points

Michael Dewey

[LINK]

Requests from Editors:

GENERAL

* Please ensure that your manuscript contains line and page numbers.

* Please confirm that your title complies with PLOS Medicine's style. Your title must be nondeclarative and not a question. It should begin with main concept if possible. "Effect of" should be used only if causality can be inferred, i.e., for an RCT. Please place the study design ("A randomized controlled trial," "A retrospective study," "A modelling study," etc.) in the subtitle (ie, after a colon).

* Your study is observational and therefore causality cannot be inferred. Please remove language that implies causality, such as effect. Refer to associations instead.

* Please ensure that all abbreviations are defined at first use throughout the text (including statistical abbreviations). Please also check figures and tables.

* Please review your text for claims of novelty or primacy (e.g. 'for the first time', ‘novel’) and remove this language.

* Please check that any use of statistical terms (such as trend or significant) are supported by the data, and if not please remove them.

* Statistical reporting: Please revise throughout the manuscript, including tables and figures.

a) Please report statistical information as follows to improve clarity for the reader "22% (95% CI [13%,28%]; p</=)".

b) Please separate upper and lower bounds with commas instead of hyphens as the latter can be confused with reporting of negative values.

c) Please define statistical definitions at first use and repeat the abbreviated definitions (HR, CI etc.) for each set of parentheses.

* Please revise for use of patient-centered language. Please note that patient-centered language is constructed with the use of post-modified nouns (e.g. 'patients with ischemia’ (or similar) instead of ‘ischemic patients’) putting the person first in the sentence structure.

* The terms gender and sex are not interchangeable (as discussed in https://www.who.int/health-topics/gender#tab=tab_1 ); please use the appropriate term. Since you are referring to sex, please use the terms ‘females/males’ instead of ‘women/men’ in your study.

* p.35: Please ensure that the information provided here matches the metadata provided in the online submission form.

* Throughout the manuscript you refer to 'at 90 days' without specifying what this time point refers to (e.g. 90 days post-intervention). Please clarify and revise throughout.

* Please clarify throughout the manuscript that you are comparing endovascular thrombectomy plus medical care to medical care alone (you often write endovascular thrombectomy compared to medical care alone).

* Please consider if moving some of the figures shared in the supplementary information into the main text would aid the reader.

DATA AVAILABILITY

* Please clarify what you mean by “we upload the data for outcome calculation”. Please clarify what type of data you are referring to and where the data will be uploaded (e.g. Supporting Information, repository etc.).

ABSTRACT

* Please confirm that your abstract complies with our requirements, including providing all the information relevant to this study type https://journals.plos.org/plosmedicine/s/submission-guidelines#loc-abstract

* Please ensure that all numbers presented in the abstract are present and identical to numbers presented in the main manuscript text.

* Please mention whether there were any language restriction for the search.

* Please provide the synthesis/appraisal methods.

* Please clarify why for the outcome on reduced disability you have provided minimum and maximum possible NNT (instead of 95% CI values).

* The term trend should be used only when the test for trend has been conducted. Please revise accordingly.

AUTHOR SUMMARY

* Please revise the format and insert bullet points.

* Please clarify that you compared endovascular thrombectomy plus medical care to medical care alone.

* Please clarify that the results refer to 90 days post-procedure.

INTRODUCTION

* Given that PLOS Medicine has a wide readership, we suggest that the introduction could briefly explain what endovascular thrombectomy is, but we leave you to decide.

METHODS AND RESULTS

* Please include a flow diagram as Figure 1 (according to the PRISMA guidelines).

* Please briefly mention what the modified Rankin scale measures, e.g. that it is a 6-level ordinal categorical scale outcome measure used primarily to classify the degree of disability in stroke patients.

* p.8: Do the search terms you present on page 8 reflect the full search strategy?

* p.12: Please explain the Grading of Recommendations in more detail.

* p.11: “All statistical tests for heterogeneity are weak, including I².” – please rephrase and avoid making absolute statements based on a single reference. We think it would be better to focus on the fact that I² should be interpreted with caution when a meta-analysis has few studies.

* p.14, “73%” – we suggest providing the numerator and denominator for transparency.

* p.14: Please briefly describe the results for the risk of bias assessment.

* Table 1:

a) Please define ‘y’ (years) or spell out 'y'.

b) For the category ‘Age’, please clarify whether you are presenting mean or median.

c) Please also present ranges (e.g. standard deviation or IQR) when presenting mean or median.

* Figure 1: Please define ‘IV’ and ‘CI’. Please note that in the axis title it says “Favors Meical Care” – please correct. Please clarify what ‘at 90 days’ mean (at 90 days post ?).

* Figure 2: Please correct ‘Endovascular Thrombecomy’. Please clarify that the first group entails ‘Endovascular Thrombectomy plus medical care’. Please clarify what ‘at 90 days’ mean (at 90 days post ?).

* Figure 3: Please define ‘IV’ and ‘CI’. Please clarify what ‘at 90 days’ mean (at 90 days post ?).

* p.23: “associated with a trend toward higher proportion of patients“ - the term trend should be used only when the test for trend has been conducted. Please revise accordingly.

* p.24: Please describe the results for GRADE in the main text in more detail. We suggest moving the table into the main results.

* S8 Figure: Please note that the figure seems to be minimally cut-off at the lower edge.

* p.27: Please expand the section on the funnel plot (e.g. mention that the funnel plot is used to detect publication bias) or include the results elsewhere. We do not think it’s necessary that a one liner requires its own section with its own subheading.

* The results section currently has many subheadings, which we suggest should be reduced. Some of the subheadings could instead be used to introduce a paragraph. For example, on page 21, Safety Outcomes: "For the first safety outcome of requiring constant care or death at 90 days (mRS score of 5 to 6), the pooled results showed that EVT...". Please revise throughout accordingly.

DISCUSSION

* Please remove any subheadings in the Discussion, including the Conclusion subheading.

* p.27 “less disabled outcomes” – please re-phrase and revise accordingly throughout.

* p.28, please change to: “ appears to be evidence-based and clinically justified”.

* p.31: We feel that you jump very abruptly to stating that the current study differs from the study by Chen et al. Please revise and, for example, start the paragraph by stating that a similar meta-analysis has recently been published.

* p.31, “we were unable to conduct subgroup analyses” – please clarify since you have conducted a subgroup analysis.

General Editorial Requests

---

## [Editor Report · Decision Letter 3]

25 Mar 2025

Dear Dr. Lee,

Thank you very much for re-submitting your manuscript "Association between Endovascular Thrombectomy and Outcomes in Acute Stroke with a Large Ischemic Core: A Systematic Review and Meta-analysis of Randomized Controlled Trials" (PMEDICINE-D-24-03296R3) for review by PLOS Medicine.

There are a few minor editorial issues that need to be addressed before we can accept the manuscript for publication; these are outlined at the end of this email. Please revise the paper accordingly, and submit the final revision until March 28.

Please ensure you address the specific points made by the editors. In your rebuttal letter you should indicate your response to the reviewers' and editors' comments and the changes you have made in the manuscript. Please submit a clean version of the paper as the main article file. A version with changes marked must also be uploaded as a marked up manuscript file. Please also check the guidelines for revised papers at http://journals.plos.org/plosmedicine/s/revising-your-manuscript for any that apply to your paper.

A reminder that when your manuscript is accepted, an uncorrected proof of your manuscript will be published online ahead of the final version, unless you've already opted out via the online submission form. If, for any reason, you do not want an earlier version of your manuscript published online or are unsure if you have already indicated as such, please let the journal staff know immediately at plosmedicine@plos.org.

If you have any questions in the meantime, please contact me directly at atosun@plos.org.

We look forward to receiving the revised manuscript.

Sincerely,

Alexandra Tosun, PhD

Associate Editor

PLOS Medicine

Requests from Editors:

1) Title: We suggest changing the title to: Endovascular Thrombectomy in Acute Stroke with a Large Ischemic Core: A Systematic Review and Meta-analysis of Randomized Controlled Trials

2) Abstract: Please change the search dates to “January 1 2000 to September 25 2024”.

3)Abstract: Please note that we have some flexibility on the length of the abstract, i.e. you may include the NNT number (plus minimum and maximum NNT) for functional outcomes/reduced disability, and we encourage you to do so.

4) Abstract: Given that you have described the primary secondary outcome as reduced disability, we feel that the description of "better functional outcomes" later on in the Abstract may be confusing to readers. Please revise and use the same description throughout the manuscript. You seem to have changed "reduced disability" to "better functional outcomes" throughout, so we would prefer to stick with the latter.

5) Introduction: Thank for including a brief introduction on endovascular thrombectomy. Please provide a reference.

6) Figure 1: In the second box on the right, please change the text to “Records excluded due to duplicate”.

7) Table 2: Please define ‘mRS’ and replace commas with hyphens in the effect column.

8) Please note that there are still instances in tables (e.g. Table 4) where you report values, e.g. confidence intervals, with commas instead of hyphens. Please carefully check and revise throughout.

9) Please ensure to upload the completed PRISMA checklist.

10) Table 1: Please change "men±SD" to "mean±SD" (fourth row, age).

---

## [Editor Report · Decision Letter 4]

31 Mar 2025

Dear Dr Lee, 

On behalf of my colleagues and the Guest Academic Editor, Iris Grunwald, I am pleased to inform you that we have agreed to publish your manuscript "Endovascular Thrombectomy in Acute Stroke with a Large Ischemic Core: A Systematic Review and Meta-analysis of Randomized Controlled Trials" (PMEDICINE-D-24-03296R4) in PLOS Medicine.

I appreciate your thorough responses to the reviewers' and editors' comments throughout the editorial process.

PRESS

Sincerely, 

Alexandra Tosun, PhD 

Associate Editor 

PLOS Medicine